# Prediction Potential of Remote Sensing-Related Variables in the Topsoil Organic Carbon Density of Liaohekou Coastal Wetlands, Northeast China

**Shuai Wang** [1,2,†], **Mingyi Zhou** [2,†], **Qianlai Zhuang** [3] and **Liping Guo** [1,*]

1. Key Lab for Agro-Environment, Ministry of Agriculture and Rural Affairs, Institute of Environment and Sustainable Development in Agriculture, China Academy of Agricultural Science, Beijing 100081, China; shuaiwang666@syau.edu.cn
2. College of Land and Environment, Shenyang Agricultural University, Shenyang 110866, China; 2018156015@stu.syau.edu.cn
3. Department of Earth, Atmospheric, and Planetary Sciences, Purdue University, West Lafayette, IN 47907, USA; qzhuang@purdue.edu
* Correspondence: GuoLiping@caas.cn; Tel.: +86-10-8210-6045
† These authors contributed equally to this paper.

**Abstract:** Wetland ecosystems contain large amounts of soil organic carbon. Their natural environment is often both at the junction of land and water with good conditions for carbon sequestration. Therefore, the study of accurate prediction of soil organic carbon (SOC) density in coastal wetland ecosystems of flat terrain areas is the key to understanding their carbon cycling. This study used remote sensing data to study SOC density potentials of coastal wetland ecosystems in Northeast China. Eleven environmental variables including normalized difference vegetation index (NDVI), difference vegetation index (DVI), soil adjusted vegetation index (SAVI), renormalization difference vegetation index (RDVI), ratio vegetation index (RVI), topographic wetness index (TWI), elevation, slope aspect (SA), slope gradient (SG), mean annual temperature (MAT), and mean annual precipitation (MAP) were selected to predict SOC density. A total of 193 soil samples (0–30 cm) were divided into two parts, 70% of the sampling sites data were used to construct the boosted regression tree (BRT) model containing three different combinations of environmental variables, and the remaining 30% were used to test the predictive performance of the model. The results show that the full variable model is better than the other two models. Adding remote sensing-related variables significantly improved the model prediction. This study revealed that SAVI, NDVI and DVI were the main environmental factors affecting the spatial variation of topsoil SOC density of coastal wetlands in flat terrain areas. The mean (±SD) SOC density of full variable models was 18.78 (±1.95) kg m$^{-2}$, which gradually decreased from northeast to southwest. We suggest that remote sensing-related environmental variables should be selected as the main environmental variables when predicting topsoil SOC density of coastal wetland ecosystems in flat terrain areas. Accurate prediction of topsoil SOC density distribution will help to formulate soil management policies and enhance soil carbon sequestration.

**Keywords:** remote sensing; soil organic carbon; coastal wetland; digital soil mapping; boosted regression tree

## 1. Introduction

Coastal wetlands are a unique ecosystem between terrestrial and marine ecosystems, which plays an irreplaceable role in maintaining biodiversity, adjusting and improving climate and providing habitat. Because of their unique ecosystem function, they are known as the kidneys of the Earth [1–3]. Coastal wetlands generally have high primary productivity. Their surface is often submerged by water, resulting in poor sediment ventilation and low surface temperature compared with unflooded surfaces, which is conducive to the preservation of organic matter and often leads to the accumulation of a

large amount of organic carbon [3,4]. In estuarine areas, their accumulated organic carbon is occasionally buried by minerals brought by river flooding, so that the accumulated organic carbon of coastal wetlands is usually higher than that of other freshwater wetlands in the world [2]. Therefore, coastal wetlands play an important role in the global carbon cycle and are considered to be one of the main burial areas of organic carbon [5–7].

In recent years, many studies have shown that digital soil mapping (DSM) can evaluate the spatial and dynamic changes of soil organic carbon (SOC), which is helpful to analyze and understand the carbon cycle in ecosystems [8–10]. However, the quantitative prediction mapping of SOC usually needs a large number of measured sample data and multi-source environmental variable data [9,11]. Therefore, SOC mapping is often limited by the cost and accuracy of data acquisition. Because of its low cost and wide spatial coverage, remote sensing technology has gradually become a key means to obtain the spatial distribution of SOC [10,12]. It is difficult to accurately predict the spatial distribution of SOC density through a single terrain variable, particularly in coastal wetland areas with flat terrain [5,7,11]. The rapid development of satellite remote sensing has enabled large-scale SOC density mapping in coastal wetland areas with flat terrain.

For coastal wetlands with wide area and inconvenient sampling, traditional large-scale data collection is undoubtedly expensive, time-consuming, and unrealistic. It is necessary to combine classical soil survey with advanced remote sensing technology to study their spatial distribution of SOC. Satellite remote sensing technology has proven to be efficient at developing data regarding the spatial distribution of soil [13–17]. The quantitative relationship between reflectance spectra and SOC has long been revealed [18,19]. There are many available remote sensing information sources at various resolutions, such as IKONOS with 4 m resolution [20], SPOT with 10 m and 20 m resolution [19], or Landsat TM/ETM with 15 m and 30 m resolution [10,18]. In addition, there are many quantitative analysis techniques, such as multivariate linear regression models (MLR) [18], partial least squares regression (PLSR) [21], artificial neural network (ANN) [22], or boosted regression tree (BRT) [10].

Different from other DSM methods, BRT is a model developed by tree-based algorithm [10], which can combine multiple simple tree models into a tree model with better performance [23]. In addition, BRT relies on stochastic gradient propulsion technology, which can make the model more accurate and faster through numerical optimization and regularization. Therefore, the BRT model can deal with linear, nonlinear, polynomial, and other problems flexibly, and the over-fitting and merging of the model can effectively improve the prediction performance of the model [23]. Based on the above advantages, BRT is widely used in soil science [24], ecology [25], medicine [26], and remote sensing science [24].

In this study, DSM technology and remote sensing data were combined to predict the topsoil SOC density in the coastal wetland area with flat terrain of Liaohe Delta. The specific research objectives included:

(1) To construct the best model for predicting topsoil SOC density of coastal wetlands based on 193 sample data and 9 environmental variables;
(2) To discuss the importance of using remote sensing data in predicting topsoil SOC density of coastal wetlands;
(3) To analyze the uncertainty of our method and results.

## 2. Materials and Methods

### 2.1. Description of the Study Area

Liaohekou coastal wetlands (40.65°–41.62° N, 121.42°–121.52° E) is located in the Southwest of Liaoning Province, Northeast China, with a total of 4071 km$^2$ (Figure 1). It is the largest coastal reed swamp in the Chinese high latitude area, with a large area of Suaeda salsa, tidal flat and shallow sea area, which is a very sensitive wetland ecosystem. This region has a warm temperate, continental, sub-humid monsoon climate, with an average annual temperature of about 9.2 °C and an average annual precipitation of 651 mm. It

is characterized by four distinct seasons, the same period of rain and heat, appropriate temperature, and sufficient illumination. There are 21 rivers in the study area, with a drainage area of 3570 km². The terrain is flat, and the average ground elevation is about 4 m. The flora are North China flora, which have fast growth and a high density of wild plants. There are 242 species of wild plants in 70 families. Among them, there are 119 species in 33 families of forest trees, while there are 123 species of weeds in 37 families. In this study, the dominant soil types are Luvisols (16%), Cambisols (33%), Gleysols (14%), Solonchaks (26%), and Leptisols (11%) according to the classification of the World Reference Base for Soil Resources (WRB) [27].

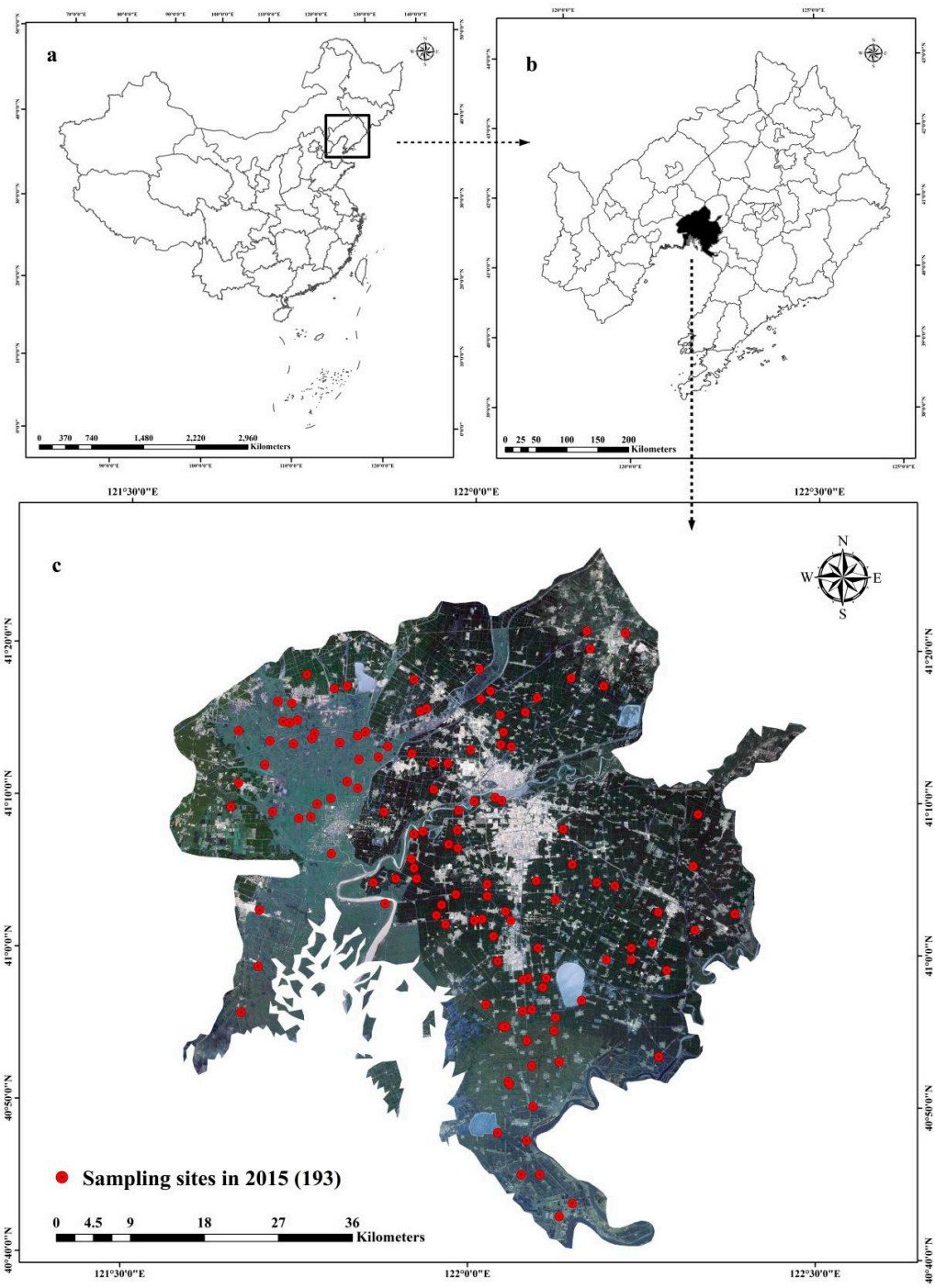

**Figure 1.** Soil sample locations overlaid on the remote sensing map of the study area (**c**) in Liaoning Province (**b**) of China (**a**).

*2.2. Soil Sampling and Laboratory Analysis*

In order to reduce cost and time, we chose a purposive sampling method to design the sampling scheme of this study [28]. This method is based on the soil landscape model and assumes that the soil has high similarity under similar environmental conditions [29,30]. First, the climate, terrain and biological factors covering the whole study area were clustered by a fuzzy C-means clustering method, and a total of 38 clustering units were obtained. Second, in each cluster unit, experts familiar with local conditions were invited to select 5–10 representative points for soil sample collection according to different terrain and road accessibility, and the point information was recorded by a handheld GPS. Finally, we collected about 1 kg soil samples at each selected point, and a total of 193 topsoil (0–30 cm) samples were obtained. The sampling points are shown in Figure 1. In the Central Laboratory of Shenyang Agricultural University, we dried the soil samples naturally, removed litter, ground them, and passed them through a 2 mm diameter nylon sieve and measured for SOC content using a wet oxidation method (Walkley–Black method) [31]. In order to estimate the dry bulk density, a 100 cm$^3$ undisturbed core was collected from the topsoil and dried at 105 °C for 48 h for bulk density measurement.

*2.3. Calculation of SOC Density*

This study analyzed the spatial distribution of SOC density in Liaohekou coastal wetlands. For a single profile with K layers (within the first meter), Batjes [32] equation was used to calculate the density of soil organic carbon (SOC) in the whole soil profile:

$$SOC \quad density = \sum_{i=1}^{k} SOC_{content} = \sum_{i=1}^{k} SOC_{concentration} \times BD_i \times D_i \times (1 - S_i) \qquad (1)$$

where *SOC density*, *SOC$_{content}$* and *SOC$_{concentration}$* are SOC density of whole soil profile (kg m$^{-2}$), SOC content and SOC concentration; *i* represents a specific soil layer; *BD$_i$* (g cm$^{-3}$), *D$_i$* (m) and *S$_i$* are the bulk density, the soil thickness, and the volume fraction of fragments'> 2 mm, respectively.

*2.4. Environmental Variables*

In this study, the environmental variables involved include two categories of remote sensing-related variables and topographic variables, a total of 11 environmental variables, namely normalized difference vegetation index (NDVI), difference vegetation index (DVI), soil adjusted vegetation index (SAVI), renormalization difference vegetation index (RDVI), ratio vegetation index (RVI), topographic wetness index (TWI), elevation, slope aspect (SA), slope gradient (SG), mean annual temperature (MAT), and mean annual precipitation (MAP) were selected. Since these variables come from different platforms and have different resolutions, these data were resampled to 30 m × 30 m spatial resolution in ArcGIS 10.2, and the projection system was unified to "krasovsky-1940-albers" for subsequent modeling.

2.4.1. Remote Sensing Related Variables

Five remote sensing-related environmental variables were derived from Landsat 8 satellite. The data (four images) were downloaded from the United States Geological Survey (USGS) (accessed on 2 March 2020, https://www.usgs.gov) between July to September in 2015 with cloud cover less than 10%. The spatial resolution of remote sensing data is 30 m and the data level was L1T, which had been corrected for geometric accuracy. Therefore, there was no need to use ground control points or digital elevation model (DEM) data for geometric accuracy correction again. The following steps were taken to process the data: Firstly, the homomorphic filtering method [33] was selected to remove the cloud from remote sensing data in MATLAB software; secondly, previous studies had revealed that bottom of atmosphere (BOA) reflectance was the most appropriate remote sensing-related variable to predict the spatial distribution of SOC density [10]. However, in order to obtain the BOA reflectance, the remote sensing data should be atmospheric corrected. In this study, we used the Fast Line-of-sight Atmospheric Analysis of Spectral Hypercubes (FLAASH) at-

mospheric correction method [34] to calibrate the atmospheric in ENVI 5.1 software; thirdly, due to the height angle of the sun, some remote sensing images seem to have mountain shadow, so we used the ratio method [35] in ENVI 5.1 software to eliminate it. In addition, since the study area was plain, we did not carry out topographic correction. Finally, the region of interest (ROI) (the boundary vector layer of Liaohekou coastal wetlands obtained from the Department of Natural Resources of Liaoning Province, China) was used to crop the remote sensing image data in ENVI 5.1. Ultimately, we could obtain remote sensing image data covering different bands of the whole study area for calculating remote sensing related variables. Among all vegetation indices, NDVI is recognized as the best index to characterize plant growth and is widely used in the prediction of crop yield and growth status [24,36]. Compared with NDVI, SAVI is able to add the soil regulation coefficient "L", determined according to the actual situation, with a value range of 0–1 [37]. When L is 0, it means that the vegetation coverage is zero; When L is 1, the vegetation coverage is very high which means that the influence of soil background is zero. This situation can only occur where there is sufficient coverage of tall trees with dense canopy. Huete's study was conducted in areas with good vegetation [37]. It was found that SAVI can better eliminate the influence of soil reflectance when the L coefficient of soil regulation is 0.5. RDVI, can better identify water bodies, and that its value will increase with the increase of vegetation coverage [38]. Previous studies have shown that, especially in the case of medium or low vegetation coverage. RVI has high performance in areas with strong vegetation growth and high coverage, which can better reflect the difference of vegetation coverage and growth status [39]. The calculation formula of these indicators is as follows:

$$SAVI = (B_{NIR} - B_{RED})(1 + L)/(B_{NIR} + B_{RED} + L) \; L = 0.5 \tag{2}$$

$$NDVI = (B_{NIR} - B_{RED})/(B_{NIR} + B_{RED}) \tag{3}$$

$$RVI = B_{NIR}/B_{RED} \tag{4}$$

$$DVI = B_{NIR} - B_{RED} \tag{5}$$

$$RDVI = \sqrt{NDVI \times DVI} \tag{6}$$

where $B_{NIR}$ and $B_{RED}$ represent Landsat 8 OLI near-infrared band and red band, respectively; $L$ represents the soil adjustment coefficient, and its value range is 0–1. In Huete's study, they found that the best setting is 0.5 in wetland areas. In this study, we set it to 0.5.

### 2.4.2. Topographic Variables

Topography is the most widely used environmental factor among the five soil forming factors, and is widely used in the research of spatial prediction of SOC density [10,17,40]. In this study, we selected four topographic variables including elevation, slope aspect (SA), slope gradient (SG) and topographic wetness index (TWI), which were derived from the 30 m spatial resolution digital elevation model (DEM) downloaded from the geospatial data cloud site (accessed on 7 October 2020, http://www.gscloud.cn). Elevation, SG and SA were derived by ArcGIS 10.2 software, and TWI was derived using SAGA GIS [41].

### 2.4.3. Climatic Variables

Climatic variables involved in this study were downloaded from the Resource and Environmental Science and Data Center of the Chinese Academy of Sciences (accessed on 18 January 2020, https://www.resdc.cn/Default.aspx). In this study, MAT and MAP were selected to represent climate factors. The data were generated through sorting, calculation and spatial interpolation based on the daily observation data of more than 2400 meteorological stations in China since 1980. The interpolation application is ANUSPLIN interpolation software from Australia. ANUSPLIN is a tool for analyzing and interpolating multivariable data by using smooth spline function, that is, a method of using function to approximate surface [42]. It can carry out reasonable statistical analysis and data diagnosis, analyze the spatial distribution of data, and then realize the function of spatial interpolation. The

downloaded data are a 1-km grid layer, which was reduced to 30 m grid in ArcGIS 10.2 for research.

## 2.5. Prediction Model

The boosted regression tree (BRT) was selected as the prediction model for SOC density of Liaohekou coastal wetlands in northeast China. The BRT model was proposed by Friedman et al. [43], who improved the performance of the model by training and combining multiple models. This model consists of two algorithms: a regression tree and gradient [23]. After several rounds of parameterization, the parameter error of the model is minimized, and the final BRT model is obtained to predict the unknown area. We use "dismo" R package version 0.8–17 [44] to conduct the BRT model in an R language environment [45]. In the BRT model the user needs to set a four parameters-learning rate (LR), tree complexity (TC), bag fraction (BF) and tree number (NT). LR represents the contribution of each tree to the final fitting [26]. TC is the complexity of the tree and the maximum level of interaction between predictors [10]. BF represents the proportion of data used in the modeling dataset [46]. Although the BRT model can avoid over fitting by extending the operation of the model, it is still necessary to set NT. NT can be determined by the combination of LR and TC [24]. A 10-fold cross-validation was used to optimize the model parameters to obtain the best prediction. The optimal values of LR, TC, BF and NT were 0.025, 9, 0.55 and 1200, respectively

## 2.6. Model Validation

In order to obtain the best prediction model of coastal plain wetland area, we made different combinations of topographic and remote sensing related variables to build the model (only topographic variable, MA; only topographic and climatic variable, MB; topographic, climatic, and remote sensing related variables, MC). Each model was iterated 100 times and their average prediction results were calculated as the final prediction results to ensure the stability of the model prediction. We randomly divided the data into two parts, 70% sampling sites data (135) for constructing the model and the other 30% sampling sites data (58) for testing the performance of the model. Four popular indicators: mean absolute error (MAE), root mean square error (RMSE), coefficient of determination ($R^2$) and Lin's consistency correlation coefficient (LCCC) [47] to evaluate the prediction performance of the three BRT model. The specific calculation formula is as follows:

$$MAE = \frac{1}{n} \sum_{i=1}^{n} |P_i - O_i| \tag{7}$$

$$RMSE = \sqrt{\frac{1}{n} \sum_{i=1}^{n} (P_i - O_i)^2} \tag{8}$$

$$R^2 = \frac{\sum_{i=1}^{n} (P_i - \overline{O}_i)^2}{\sum_{i=1}^{n} (O_i - O_i)^2} \tag{9}$$

$$LCCC = \frac{2r \partial_P \partial_O}{\partial_P^2 + \partial_O^2 + (\overline{P} + \overline{O})^2} \tag{10}$$

where $P$, $O$, $\overline{P}$, $\overline{O}$ represent the predicted value, observed value, average value of predicted and average value of observed, respectively; $i$ represents the sampling point; $\partial_P$ and $\partial_O$ represent the change between the predicted value and the observed value, respectively; $n$ represents the number of sampling points; and $r$ represents the Pearson correlation coefficient.

## 3. Results

### 3.1. Descriptive Statistics

Table 1 presented the descriptive statistics of SOC density in topsoil (0–30 cm) of Liaohekou coastal wetlands, as well as the environmental variables at the sampling sites. SOC density ranged from 0.33 to 28.31 kg m$^{-2}$, with an average of 11.14 kg m$^{-2}$ ($\pm$0.47). The skewness coefficient was 0.56 kg m$^{-2}$, indicating that SOC density had a slightly skewed distribution. Therefore, we conducted Ln-transformation for SOC density data for future modeling. Pearson correlation analysis between observed Ln-transformation SOC density (0–30 cm) with environmental variables sample sites is shown in Table 2. SAVI (r = 0.31), NDVI (r = 0.43), DVI (r = 0.21), and elevation (r = 0.19) were positively correlated with SOC density. Accordingly, SOC stock was negatively correlated with TWI (r = −0.23), RDVI (r = −0.29), and RVI (r = 0.16). Surprisingly, SOC density was significantly correlated with the selected remote sensing-related variables, while climatic variables (MAT and MAP), considered as efficient predictors, showed no significant results with SOC density in this study.

**Table 1.** Descriptive statistics of soil organic carbon (SOC) density (kg m$^{-2}$) (0–30 cm) with environment variables at sample sites.

| Property | Unit | Min. | Mean | Max. | SD | Skewness | Kurtosis |
|---|---|---|---|---|---|---|---|
| SOC density | kg m$^{-2}$ | 0.33 | 11.14 | 28.31 | 0.47 | 0.56 | 2.43 |
| SAVI | | 0.07 | 0.14 | 0.35 | 0.27 | −0.47 | 1.61 |
| NDVI | | 0.02 | 0.08 | 0.33 | 0.04 | −0.31 | 1.73 |
| RVI | | 0.36 | 0.76 | 1.12 | 0.34 | −1.24 | 1.22 |
| DVI | | 12.76 | 39.43 | 57.23 | 7.72 | 0.39 | 2.33 |
| RDVI | | 21.17 | 42.05 | 60.09 | 8.10 | 0.53 | 3.27 |
| Elevation | m | 1 | 2.87 | 12 | 1.12 | 1.13 | 3.24 |
| SG | degree | 0 | 0.05 | 2.16 | 0.11 | 0.93 | 0.32 |
| SA | degree | 0 | 174.62 | 360 | 92.65 | −0.42 | −0.89 |
| TWI | | 7.30 | 10.70 | 10.95 | 0.54 | 0.96 | 1.12 |
| MAT | degree Celsius | 9.33 | 9.53 | 9.67 | 0.38 | −1.21 | 2.16 |
| MAP | mm | 648.7 | 650.6 | 652.3 | 1.31 | 0.84 | 0.93 |

Note: Min., minimum; Max., maximum; SD, standard deviation.

**Table 2.** Pearson correlation analysis between observed Ln-transformation SOC density (0–30 cm) with environmental variables sample sites.

| Property | SOC Density | Elevation | SG | SA | TWI | SAVI | NDVI | RVI | DVI | MAT |
|---|---|---|---|---|---|---|---|---|---|---|
| Elevation | 0.19 * | | | | | | | | | |
| SG | 0.13 | 0.33 ** | | | | | | | | |
| SA | 0.15 | 0.15 | 0.27 ** | | | | | | | |
| TWI | −0.23 ** | −0.23 ** | −0.40 ** | −0.22 ** | | | | | | |
| SAVI | 0.31 ** | 0.19 * | 0.07 | 0.13 | −0.07 | | | | | |
| NDVI | 0.43 ** | 0.21 * | 0.08 | 0.09 | −0.05 | 0.42 ** | | | | |
| RVI | −0.16 * | −0.16 * | −0.11 | −0.14 | 0.15 | −0.17 ** | −0.28 ** | | | |
| DVI | 0.21 * | 0.28 ** | 0.17 | 0.17 | −0.08 | 0.17 * | 0.26 ** | −0.32 ** | | |
| RDVI | −0.29 ** | −0.25 * | −0.09 | −0.13 | 0.06 | 0.33 ** | 0.37 ** | −0.05 | 0.36 ** | |
| MAT | 0.09 | 0.12 | −0.06 | −0.07 | 0.13 * | −0.15 * | −0.10 | 0.06 | 0.13 | |
| MAP | 0.11 | −0.13 * | 0.08 | 0.06 | −0.12 * | 0.17 * | 0.19 * | −0.09 | 0.16 | 0.22 ** |

Note: $p < 0.05$ shown in "*"; $p < 0.01$ shown in "**".

### 3.2. Model Performance and Uncertainty

To accurately predict the topsoil SOC density of Liaohekou coastal wetlands, we constructed three BRT models composed of different variables. The predicted performance of the three BRT models is showed in Table 3. Compared with MA and MB, the MC model presented the best prediction performance, which could explain 57% of the spatial variation of SOC density in this region. The results show that the MC model had lower MAE and RMSE, but higher $R^2$ and LCCC than the MA and MB models. In order to further evaluate the performance and uncertainty of the model, we also calculated a coefficient of variation (CV) map of 100 iterations of the three BRT models. As can be seen from Figure 2, three

models produced lower CV. The average CV of MA, MB and MC were 2.81%, 3.00% and 3.14%, respectively.

**Table 3.** Summary statistics of the predictive performance of three BRT in the prediction of SOC density (kg m$^{-2}$) based on testing set.

| Model | Index | Min. | Median | Mean | Max. |
|-------|-------|------|--------|------|------|
| MA | MAE | 1.38 | 1.39 | 1.40 | 1.41 |
|    | RMSE | 1.51 | 1.52 | 1.52 | 1.54 |
|    | $R^2$ | 0.24 | 0.26 | 0.27 | 0.31 |
|    | LCCC | 0.33 | 0.34 | 0.34 | 0.35 |
| MB | MAE | 1.27 | 1.31 | 1.32 | 1.33 |
|    | RMSE | 1.47 | 1.48 | 1.49 | 1.50 |
|    | $R^2$ | 0.31 | 0.33 | 0.34 | 0.36 |
|    | LCCC | 0.37 | 0.38 | 0.39 | 0.40 |
| MC | MAE | 0.87 | 0.88 | 0.89 | 0.91 |
|    | RMSE | 0.96 | 0.97 | 0.97 | 0.98 |
|    | $R^2$ | 0.53 | 0.55 | 0.57 | 0.59 |
|    | LCCC | 0.53 | 0.54 | 0.55 | 0.57 |

Note: MA, included only topography variables; MB, included all predictors (topography and climate variables); MC, included all predictors (topography, climate, and remote sensing related variables).

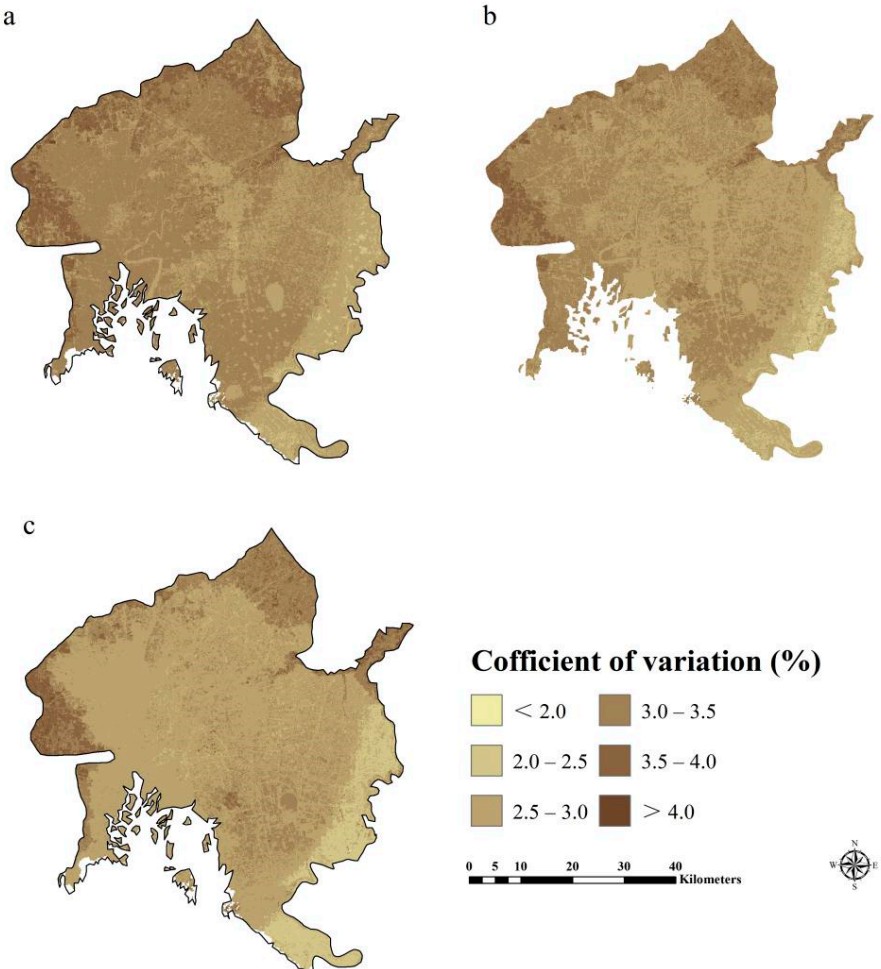

**Figure 2.** Coefficient of variation maps of predicted SOC density derived from 100 runs of the boosted regression trees (BRT). (**a**) MA model included only topography variables; (**b**) MB model included only topography and climatic variables; and (**c**) MC model included all predictors (topography, climatic and remote sensing related variables).

### 3.3. Importance of Environmental Variables

In order to obtain the relative importance (RI) of each environmental variable in predicting the topsoil SOC density in coastal plain wetland area, we iterated the three BRT models 100 times and calculated the average RI of each environmental variable. To make each variable comparable, we standardized it to 100%. In Figure 3c, the environment variables show different RI in the MB model. Among the nine environmental variables, NDVI (16.36%), SAVI (19.32%), DVI (15.37%), RDVI (11.41%), RVI (9.41%), and TWI (7.56) were the main environmental variables affecting the spatial variation of SOC density. In the MC model, we found that remote sensing-related environmental variables played a more important role, with RI accounting for about 71.87%, while the corresponding topographic and climatic variables account for 19.93% and 8.20%, respectively.

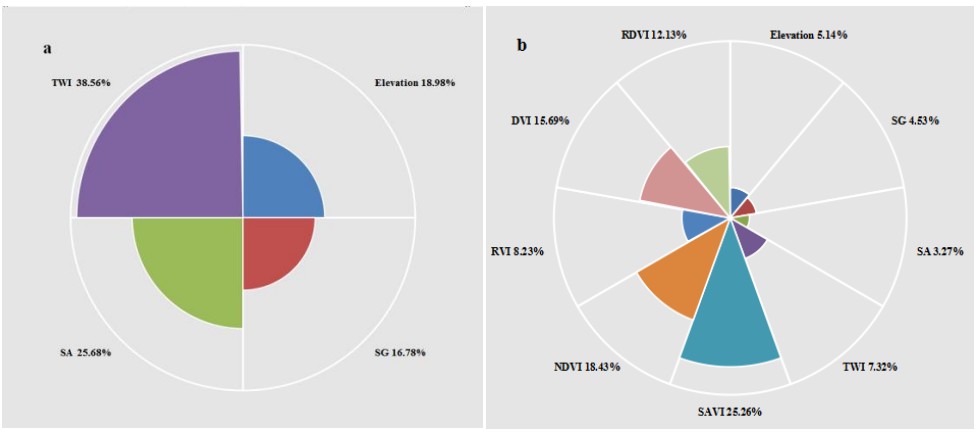

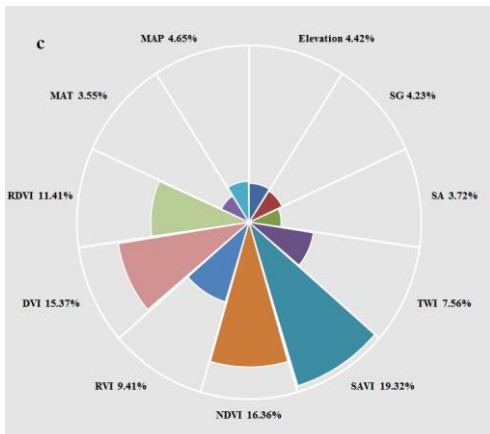

**Figure 3.** Relative importance of each variable as determined from 100 runs of the three BRT models in SOC density. (**a**) MA model included only topography variables; (**b**) MB model included only topography and climatic variables; and (**c**) MC model included all predictors (topography, climatic and remote sensing related variables).

### 3.4. Spatial Prediction of SOC Density

MA, MB, and MC models were constructed to predict the spatial distribution of topsoil (0–30 cm) SOC density in Liaohekou coastal wetlands, and the prediction maps are shown in Figure 4. The mean (±SD) SOC density of MA, MB, and MC models were 17.87(±2.35) kg m$^{-2}$, 17.97(±1.87) kg m$^{-2}$ and 18.41(±1.94) kg m$^{-2}$, respectively. The distribution characteristics of SOC density in the three maps were similar, and gradually decreased from northeast to southwest. In order to contrast the difference between MB and MC, we generated a difference map of SOC density predicted by the two models (Figure 5). The average SOC density difference between maps was 0.44 kg m$^{-2}$ and the predicted SOC density of MC model was lower than that of MB model in most study areas.

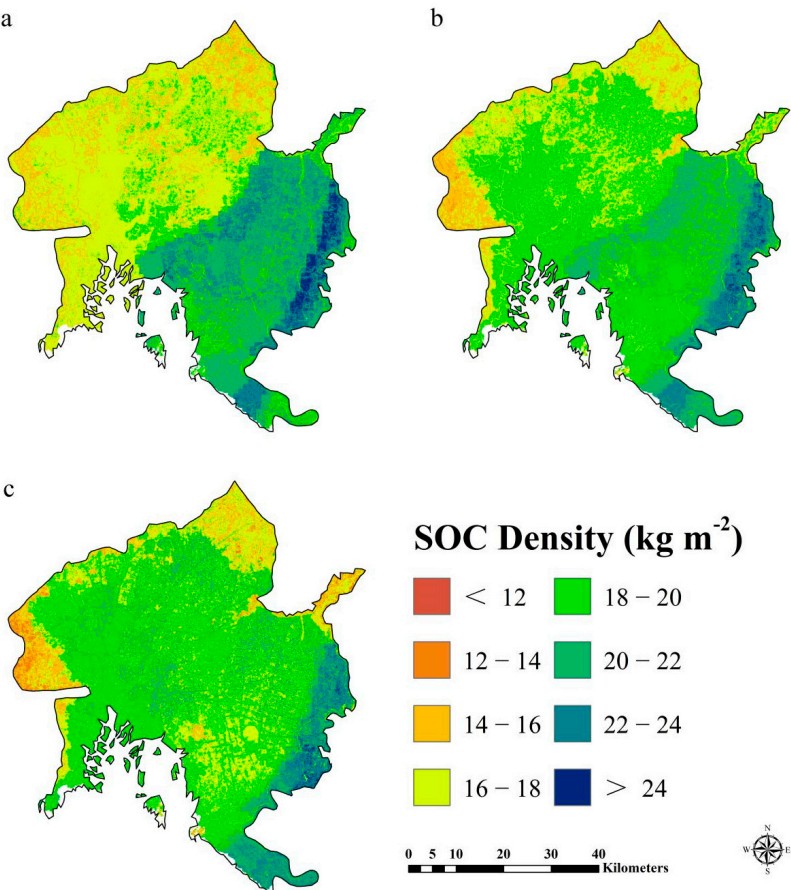

**Figure 4.** Distribution maps of predicted SOC density derived from 100 iterates of three BRT models. (**a**) MA model included only topography variables; (**b**) MB model included only topography and climatic variables; and (**c**) MC model included all predictors (topography, climatic and remote sensing related variables).

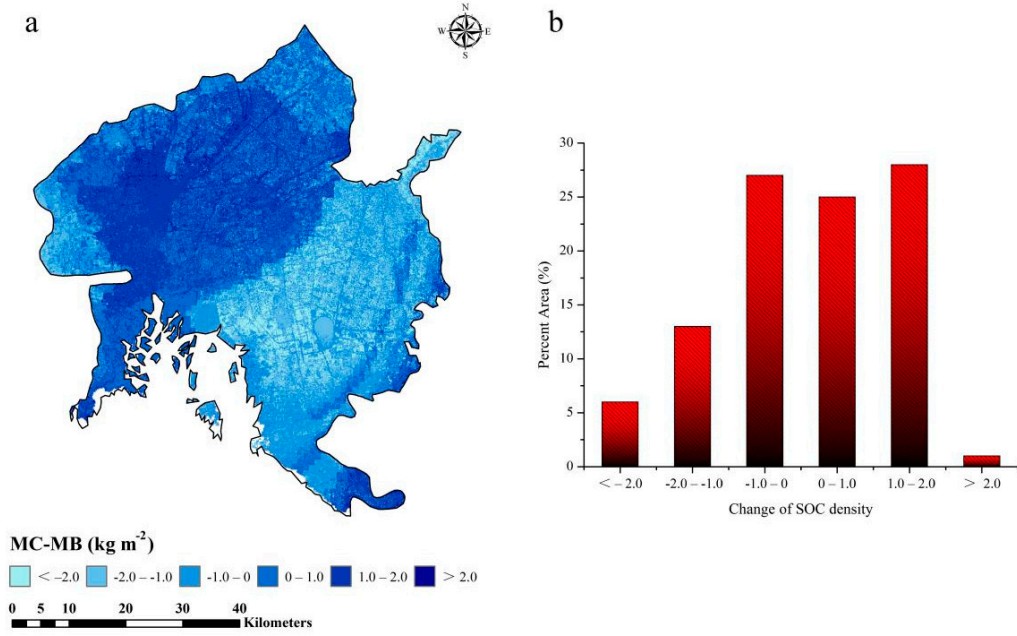

**Figure 5.** Difference map (**a**) and change (**b**) in SOC density derived from MC and MB models.

## 4. Discussion

### 4.1. Importance of Remote Sensing-Related Variables in Predicting SOC Density

In coastal plain wetland ecosystems, adding remote sensing related data could better predict topsoil SOC density (Table 3), which would be consistent with previous research [21,48–51]. In the Ebinur Lake Wetland National Nature Reserve of northwest China, Wang et al. [50] used a fractional derivative algorithm, the subsection of spectral band method, and the optimal remote sensing index to accurately predict soil organic matter content. They concluded that remote sensing-related variables helped predict soil organic matter in the topsoil of wetlands. Previous studies had also found that vegetation was associated with topsoil SOC. Kim and Grunwald [49] developed a random forest model to predict SOC density using field observation data, environmental ancillary data, and spectral data derived from remote sensing images. They concluded that vegetation indices should be major predictors in soil carbon models. The spectral reflectance of remote sensing data and the derived vegetation index reflect the vegetation coverage that is strongly related to SOC density.

In this study, we found that NDVI, DVI and SAVI were the most powerful environmental variables affecting the variation of SOC density (Figure 3). Similar conclusions have been obtained in previous studies [7,24,27,52]. Gomez et al. [21] selected four models (cubism model, generalized linear model, support vector machine and random forest) and collected 8227 soil profiles to map SOC stock in Brazil, and pointed out that NDVI was an effective environmental variable. In a forest ecosystem of northeast China, Wang et al. [24] selected 9 remote sensing related environmental variables (NDVI, SAVI, DVI, RVI, RDVI, $B_{NIR}$, $B_{RED}$, and $B_{BLUE}$) combined with a BRT model to predict the topsoil SOC density. Qi et al. [27] concluded SAVI and NDVI should be important remote sensing variables to predict topsoil SOC density in the coastal forest region of northeast China. Jobbágy and Jackson [53] insisted that SOC presents different distribution patterns under different vegetation types. In addition, NDVI could also reflect vegetation productivity and biomass, which has been widely used in previous spatial predictions of SOC density. In MC (full variables) model, SAVI was the most important variable, and its RI was 19.23%. This finding was similar to previous studies [24,48,51]. These studies showed that SAVI was one of the main variables. Wang et al. [24] considered that SAVI was the main environmental variable controlling the change of SOC density. Because it was sensitive to the change of soil background, DVI could better identify the water body, and its value could change with the change of vegetation density. RVI was widely used in relevant research in densely vegetated areas due to it potentially being able to better counter the difference of vegetation coverage and growth. Therefore, DVI and RVI were introduced into the study of SOC density mapping in coastal plain wetland ecosystem.

This study showed that combining remote sensing-related variables with other environmental variables as SOC density predictors could significantly improve the prediction of topsoil SOC density in coastal wetlands. We found that remote sensing related variables should be considered in future SOC density modeling studies, especially in coastal plain wetlands areas.

### 4.2. SOC Distribution and Associated Predictors

The spatial variation of SOC content predicted by three BRT models composed of different combinations of environmental variables showed a comparable spatial distribution pattern or trend (Figure 4). Overall, SOC density showed a downward trend from southeast to northwest. The highest SOC density in the southern coastal area of the study area was mainly due to the main vegetation type of reed. The geographical location was in the north of China where winter is cold and long, causing the high input and low output of SOC in this area [10]. In the southeast region, with the increase of cultivated land reclamation time and human activities, the organic carbon loss rate in the region is accelerating rapidly, which has been confirmed by many previous studies [10,24].

Topographic variables were widely used in the prediction of SOC density, especially in the areas with large terrain changes [9,11,24,42,54]. In previous studies, elevation was closely related to the spatial distribution pattern of SOC density, especially in the mountainous terrain. It was attributed to the influence of elevation on microclimate, which indirectly affected the distribution of SOC density. However, in the Liaohekou coastal wetlands, the ground is flat, there are no mountains, and the slope is less than 2°. Therefore, the RI of topographic variables was relatively low in MB, accounting for only 19.89%. Among all topographic variables, TWI had the highest RI, followed by SA, SG, and elevation. This result was surprising, elevation showed the lowest RI, while TWI showed the highest RI. In our analysis, this was mainly because the study area was a coastal wetland with flat and dense rivers. The distribution of soil nutrients is controlled by the discharge and flow direction of water. To a certain extent, TWI could reflect and identify the rainfall runoff pattern, potential soil water content increase and ponding area, and could indirectly reflect the influence of soil leaching, erosion, deposition, decomposition, and horizontal distribution. We found that adding remote sensing-related variables into BRT models can significantly improve their prediction accuracy for SOC density in coastal wetlands.

Precipitation and temperature were considered to be important climate variables affecting the spatial variation of SOC density [10,14]. However, different from previous studies [9,10,16], we found that the spatial variation of climate variables in SOC density in this region was not obvious. Our analysis suggested that this might be due to the small area of the study area, so that climate changes across the area were not significant, and the spatial variation characteristics of SOC density variables could not be accurately captured in the area. Therefore, although climate variables were considered effective environmental factors in other studies [9,16], this was not the case in this study. However, due to the high decomposition rate of SOC and high precipitation in coastal areas, it was necessary to develop an environmental variable that can reflect the spatial variability of SOC density in coastal plain areas.

### 4.3. Uncertainty in Current Research

Although the results of this study show that the MC model (full variables) could well predict the spatial distribution of topsoil SOC density in coastal plain wetland areas, there were still some other uncertainties in this study. First, due to the tight time and heavy tasks, we divided into different groups for sample collection and sample analysis, which might have caused sampling or experimental errors. Second, we used ArcGIS 10.2 to resample environmental variables to produce 30 m spatial resolution, which might have caused data error. Thirdly, environment variables were obtained from different platforms and the accuracy and scale of data were different, leading to subsequent modeling errors. Fourth, because a large number of artificial surface layers are distributed in the study area, but we have not sampled samples in such areas, thus, there may be some biases in our estimation of the SOC density. Finally, this study estimated SOC density in the topsoil (0–30 cm) of coastal wetlands, which may underestimate the SOC density in this area.

### 5. Conclusions

In the coastal plain wetland area, due to the relatively flat terrain, the typical environmental variables do not adequately reflect the spatial variability of soils, so it is difficult to accurately estimate soil organic carbon density in this area. With advanced remote sensing information technology, this study took advantage of available high-resolution remote sensing data of the land surface including vegetation information to construct predictive models of soil carbon density. This study provides a sound method for wetland C density mapping, which shall help mapping soil C density for other coastal wetland flat areas in the future. Specifically, based on three BRT models with different combinations of environmental variables, SOC density in the topsoil (0–30 cm) of Liaohekou coastal wetlands in northeast China were predicted. Compared with the MA model excluding remote sensing variables, the MB model with remote sensing variables as SOC density

predictors significantly improves the prediction. The $R^2$ and LCCC of MB model were higher, MAE and RMSE were lower. The average SOC density predicted by MA and MB model was 18.67 ($\pm$2.49) kg m$^{-2}$ and 18.78 ($\pm$1.95) kg m$^{-2}$, respectively. Among the remote sensing variables, NDVI, DVI and SAVI were the most powerful important environmental variables affecting the distribution of topsoil SOC density in coastal plain wetland areas. These variables directly reflect the SOC footprint. Therefore, we found that remote sensing-related variables should be considered as potential predictors in future SOC density mapping, especially in coastal plain wetland areas. We believe that our SOC density map will have a positive impact on land use decision-making and agricultural management in this study area.

**Author Contributions:** S.W. and L.G. conceived the design of research ideas. M.Z. and L.G. undertook sample collection, the experiment, and data analysis. S.W. and M.Z. wrote the manuscript. Q.Z. made professional modifications to the manuscript. All authors have read and agreed to the published version of the manuscript.

**Funding:** This work was supported by the Doctoral Research Start-up Fund Project of Liaoning Province (2021-BS-136); China Postdoctoral Science Foundation (Grant No. 2019M660782), National Key R&D Program of China (2019YFC1407700), Agricultural Science and Technology Innovation Program of Chinese Academy of Agricultural Sciences, and Young scientific and Technological Talents Project of Liaoning Province (Grant No. LSNQN201910 and LSNQN201914).

**Institutional Review Board Statement:** Not applicable.

**Informed Consent Statement:** Informed consent was obtained from all subjects involved in the study.

**Data Availability Statement:** The data that support the findings of this study are available from the author upon reasonable request.

**Conflicts of Interest:** The authors declare no conflict of interest.

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
