# Peer review of "Prediction Potential of Remote Sensing-Related Variables in the Topsoil Organic Carbon Density of Liaohekou Coastal Wetlands, Northeast China"

_remotesensing, doi:10.3390/rs13204106_

Round 1
Reviewer 1 Report
The specific comments read as below.
- There are a lot of artificial surfaces in the Liaohekou coastal wetland. The sampling points do not include the type. Therefore, the result may have large uncertainties.
- Using soil organic carbon density instead of soil organic carbon stock is better in this MS.
- In the spatial prediction of soil organic carbon, the author considered the role of topography and remote sensing factors. However, in the discussion part of the article, the author also mentioned that soil type has an important influence on soil organic carbon. I think these factors should be fully considered, including climate factors such as temperature and precipitation.
- Line 23, "two BRT models" should not be abbreviated the first time.
- In Key words, it is recommended to add "boosted regression tree".
- In the introduction part, BRT models should be briefly introduced, including the scope of application and advantages, and the reasons for choosing this model.
- Line 125, the used software is ArcGIS 10.2, and here is ArcGIS 10.1?
- Line 133, the used band resolution is 30 m. Why did the author resample again?
- Line 217, “Min., minimum; Max., maximum; and SD, standard deviation;” these don’t appear in Table 2, and should be removed.
- Line 221, thee?
- Figure 2 doesn’t show the averaged SD of MA and MB well. At the same time, I don’t understand what the standard deviation maps of predicted SOC stocks mean. Can you give a detailed description?
- Other parts introduce the results of MA model and MB model. Why does this chapter only introduce the importance of MB model?
- Section 4.2 also explains the importance of remote sensing factors, which overlaps with Section 4.1 and needs to be reorganized.
Author Response
Response to reviewer 1 comments on on the manuscript remotesensing-1398759 “Prediction potential of remote sensing related variables in the topsoil organic carbon density of Liaohekou coastal wetlands, Northeast China ”
- There are a lot of artificial surfaces in the Liaohekou coastal wetland. The sampling points do not include the type. Therefore, the result may have large uncertainties.
Response: Thank you for your detailed comments. In this revision, we addressed this issue in our uncertainty analysis Section 4.3. L484-486
- Using soil organic carbon density instead of soil organic carbon stock is better in this MS.
Response: According to your comments, we revised the soil organic carbon stocks to soil organic carbon density in the manuscript. L3,18,18,33,36,37,67,97,L285...
- In the spatial prediction of soil organic carbon, the author considered the role of topography and remote sensing factors. However, in the discussion part of the article, the author also mentioned that soil type has an important influence on soil organic carbon. I think these factors should be fully considered, including climate factors such as temperature and precipitation.
Response: According to your comments, we have added mean annual precipitation and mean annual temperature, reconstructed the model, and added some discussion in this revision. L227-239, L463-473, Table 1 and 2
- Line 23, "two BRT models" should not be abbreviated the first time.
Response: We revised it in the manuscript. L25
- In Key words, it is recommended to add "boosted regression tree".
Response: We added "boosted regression tree" in keywords. L40-41
- In the introduction part, BRT models should be briefly introduced, including the scope of application and advantages, and the reasons for choosing this model.
Response: Based on your comments, we have added some notes on the BRT model. L81-89
- Line 125, the used software is ArcGIS 10.2, and here is ArcGIS 10.1?
Response: We have corrected this ambiguity. L190
- Line 133, the used band resolution is 30 m. Why did the author resample again?
Response: We have deleted this ambiguous expression. L188-189
- Line 217, “Min., minimum; Max., maximum; and SD, standard deviation;” these don’t appear in Table 2, and should be removed.
Response: Based on your comments, we have revised the manuscript. L304-305
- Line 221, thee?
Response: Here is “the”, we corrected it. L308
- Figure 2 doesn’t show the averaged SD of MA and MB well. At the same time, I don’t understand what the standard deviation maps of predicted SOC stocks mean. Can you give a detailed description?
Response: In order to avoid ambiguity, we revised figure 2 and recalculated the coefficient of variation for more intuitive analysis. Figure 2.
- Other parts introduce the results of MA model and MB model. Why does this chapter only introduce the importance of MB model?
Response: In order to avoid ambiguity, we recalculated the three models involved in the manuscript, and remade Figure 3.
- Section 4.2 also explains the importance of remote sensing factors, which overlaps with Section 4.1 and needs to be reorganized.
Response: In order to avoid ambiguity, we reorganized the manuscript in this revision. Section 4.1 and 4.2
We thank the reviewer 1 constructive comments, which significantly help improve our manuscript.

Reviewer 2 Report
Dear authors,
This manuscript performed and compared two BRT models with different combinations of topographic and remote sensing related variables to predict Soil Organic Carbon stock(SOC stocks). The paper is interesting but it needs to be improved. In my opinion, there is a lack of detail and clarity in methods and results. So there are some issues that need to be resolved before this manuscript can be considered for publication.
General comments:
- Why did the authors just chose and use vegetation indexes and terrain attributes to estimate SOC stocks when the performance of models are very poor based on the Ratio of Performance to Deviation (RPD=SD/RMSE) (model MA: RPD= 0.73/1.47=0.49; model MB: RPD= 0.52/1.07=0.48). So in my opinion, it is preferable to use other soil formation factors to develop a robust model for predicting SOC stocks. distribution map.
- The calculation of SOC stocks was not explained. The authors must introduce how did they measure bulk density. Did they use undistributed samples to calculate bulk density?
- The paper's innovation is unclear, and it does not provide a good argument for what is novel.
Minor comments:
- Line 27, please insert a space after “were”.
- Figure1, please add the grids of the coordinate system around the map.
- Please present each map after the first citation in each paragraph.
- Line 88, please replace “km2” with “km2”.
- Line 116, please explain which methods did you used for measuring SOC and bulk density. Did you measure bulk density after sieving!!!!? How!!?
- Please provide more detailed information about the level of Landsat and processing steps.
- Lines 138 to 139, The sentence is not clear and should be corrected.
- Lines 149 to 150, In Equations 3 and 4; the ratio vegetation index (RVI) and difference vegetation index (DVI) formulas are incorrect.
- In line 152, “Landsat 8 TM” is incorrect; “Landsat 8 OLI” should be used instead.
- In line 171, the "demo" R package was not being found in R, what kind of R package did you use to perform the model?
- For model validation, it is preferable to divide your dataset into the train (70%) and test data set (30%) and use cross-validation on the train set and then test the model with the test dataset.
- In line 89, the abbreviation of Lin's consistency correlation coefficient is LCCC but in equation 9 is LUCC, please use the same abbreviation.
- In lines 206 and 2015, “Relationships” is unclear, please explain what kind of correlation coefficient did you use? Spearman or Pearson?
- In line 221, please correct “Thee”.
- In lines 228 and 229, It is better to calculate the Ratio of Performance to Deviation (RPD) to show the performance of your model. However, based on your SD and RMSE, both models are very poor so how did you conclude (in line 336) that the models could well predict the spatial distribution of topsoil SOC stocks
- Why were not other soil forming factors used to improve and make robust models?
- Table 3, please replace SOC with SOC stocks in table caption and also bring the unit.
- Figure 2, It is better to show the uncertainty of your model with the coefficient of variation (CV% = (standard deviation/ mean predicted value) × 100) as a measure of the spatial uncertainty in the mean of your estimate.
- Figure 4, please add the unit to the legend of the SOC stocks distribution maps.
Author Response
Response to reviewer 2 comments on on the manuscript remotesensing-1398759 “Prediction potential of remote sensing related variables in the topsoil organic carbon density of Liaohekou coastal wetlands, Northeast China ”
Dear authors,
This manuscript performed and compared two BRT models with different combinations of topographic and remote sensing related variables to predict Soil Organic Carbon stock(SOC stocks). The paper is interesting but it needs to be improved. In my opinion, there is a lack of detail and clarity in methods and results. So there are some issues that need to be resolved before this manuscript can be considered for publication.
Response: Thank you for your insightful comments. According to your comments, we have revised the manuscript.
General comments:
- Why did the authors just chose and use vegetation indexes and terrain attributes to estimate SOC stocks when the performance of models are very poor based on the Ratio of Performance to Deviation (RPD=SD/RMSE) (model MA: RPD= 0.73/1.47=0.49; model MB: RPD= 0.52/1.07=0.48). So in my opinion, it is preferable to use other soil formation factors to develop a robust model for predicting SOC stocks. distribution map.
Response: Based on your comments, we added climate variables (annual mean temperature and annual precipitation) to make the model more robust. L227-239, L463-473, Table 1 and 2
- The calculation of SOC stocks was not explained. The authors must introduce how did they measure bulk density. Did they use undistributed samples to calculate bulk density?
Response: According to your comments, we added this part in the manuscript. Our calculation of soil organic carbon is based on the sampled points. L137-150
- The paper's innovation is unclear, and it does not provide a good argument for what is novel.
Response: Thank you for your comments. We added the following section to conclusion section “In the coastal plain wetland area, due to the relatively flat terrain, the typical environmental variables do not adequately reflect the spatial variability of soils, so it is difficult to accurately estimate soil organic carbon density in this area. With the advanced remote sensing information technology, this study took advantage of the available high-resolution remote sensing data of the land surface including vegetation information to construct predictive models of soil carbon density. This study provides a sound method for wetland C density mapping, which shall help mapping soil C density for other coastal wetland flat areas in the future” . L490-499
Minor comments:
- Line 27, please insert a space after “were”.
Response: We corrected it. L32
- Figure1, please add the grids of the coordinate system around the map.
Response: We corrected it. Figure1
- Please present each map after the first citation in each paragraph.
Response: We corrected it. Figure1,2,3,4
- Line 88, please replace “km2” with “km2”.
Response: We corrected it. L105
- Line 116, please explain which methods did you used for measuring SOC and bulk density. Did you measure bulk density after sieving!!!!? How!!?
Response: Based on your comments, we have supplemented this part. L136-150
- Please provide more detailed information about the level of Landsat and processing steps.
Response: Based on your comments, we have supplemented this part. L163-190
- Lines 138 to 139, The sentence is not clear and should be corrected.
Response: According to your comments, we revised it in this revision. L193-200
- Lines 149 to 150, In Equations 3 and 4; the ratio vegetation index (RVI) and difference vegetation index (DVI) formulas are incorrect.
Response: According to your comment, we corrected them. Equations 4 and 5
- In line 152, “Landsat 8 TM” is incorrect; “Landsat 8 OLI” should be used instead.
Response: According to your comment, we corrected it. L214
- In line 171, the "demo" R package was not being found in R, what kind of R package did you use to perform the model?
Response: Based on your comments, we corrected it. L247
- For model validation, it is preferable to divide your dataset into the train (70%) and test data set (30%) and use cross-validation on the train set and then test the model with the test dataset.
Response: According to your comments, we have modified the manuscript. L264-270,306-330
- In line 89, the abbreviation of Lin's consistency correlation coefficient is LCCC but in equation 9 is LUCC, please use the same abbreviation.
Response: According to your comment, we corrected it. Equation 10
- In lines 206 and 2015, “Relationships” is unclear, please explain what kind of correlation coefficient did you use? Spearman or Pearson?
Response: This is Pearson correlation analysis. we corrected it. L290, 302
- In line 221, please correct “Thee”.
Response: According to your comment, we corrected it. L308
- In lines 228 and 229, It is better to calculate the Ratio of Performance to Deviation (RPD) to show the performance of your model. However, based on your SD and RMSE, both models are very poor so how did you conclude (in line 336) that the models could well predict the spatial distribution of topsoil SOC stocks
Response: Thank you for your careful and patient comments. RPD is a meaningful indicator, but the SD listed in this study refers to the standard deviation map of 100 iteration prediction results, not the SD of the verification sites. RMSE refers to the RMSE of the verification sites, so this result is generated. In order to avoid ambiguity, we recalculated the coefficient of variation for analysis. Figure.2
- Why were not other soil forming factors used to improve and make robust models?
Response: According to your comments, we added climate variables (annual mean temperature and annual precipitation) to the model. L227-239, L463-473, Table 1 and 2
- Table 3, please replace SOC with SOC stocks in table caption and also bring the unit.
Response: We corrected it. Table 3
- Figure 2, It is better to show the uncertainty of your model with the coefficient of variation (CV% = (standard deviation/ mean predicted value) × 100) as a measure of the spatial uncertainty in the mean of your estimate.
Response: Based on your comments, we have modified it. Figure 2
- Figure 4, please add the unit to the legend of the SOC stocks distribution maps.
Response: Based on your comments, we revised it. Figure 4
We thank the reviewer 2 constructive comments, which significantly help improve our manuscript.

Reviewer 3 Report
Dear Authors,
The paper requires additional clarifications, which are mostly included in the comments in the document. It is important to state the method of soil properties analysis. Also it is important to explain how the carbon stock was calculated. The vegetation indices have a great impact on carbon stock, but it is necessary to clarify the relationship between measured and predicted carbon stock.
Best regards,

Author Response
Response to reviewer 3 comments on on the manuscript remotesensing-1398759 “Prediction potential of remote sensing related variables in the topsoil organic carbon density of Liaohekou coastal wetlands, Northeast China ”
The paper requires additional clarifications, which are mostly included in the comments in the document. It is important to state the method of soil properties analysis. Also it is important to explain how the carbon stock was calculated. The vegetation indices have a great impact on carbon stock, but it is necessary to clarify the relationship between measured and predicted carbon stock.
Response: Thank you for your careful and patient comments.
- L99 Cambosols change to Cambisols
Response: Based on your comments, we revised it. L117-119
- L101 It will be good to present soil classification according WRB classification.
Response: Based on your comments, we revised it. L117-119
- L115 It is necessary to specify methods for SOC content, BD, and other parameters. Also, Authors need to explain how are you calculate SOC stock.
Response: Based on your comments, we revised it. L137-150
- L158 How you calculate SOC stock?
Response: Based on your comments, we revised it. L142-150
- L176 Which data used in the analysis?
Response: This is the modeling data. L252
- L203Is this results from boosted regression tree (with data from RS - vegetation indices) or calculate from measurement value SOC content, BD, according GPG LULUCF, ICPP methodology?
Response: Here are the measured results, not the predicted results of the boosted regression tree.
- L254 units?
Response: Based on your comments, we revised it. Figure 2
- L258 units?
Response: Based on your comments, we revised it. Figure 4
- L266 But MB regression explain 54%, which is much more then MA regression.
Response: Based on your comments, we correct this ambiguous description. L369-370
- L312 Jobbagy insist on distribution of SOC under different vegetation types
Response: According to your comments, we have revised this sentence again. L475-476
We thank the reviewer 3 constructive comments, which significantly help improve our manuscript.

Round 2
Reviewer 2 Report
Dear Authors.
The research paper has been revised properly. Just, add the units for BD and D on line 134.
Author Response
Response to reviewer 2 comments on on the manuscript remotesensing-1398759 “Prediction potential of remote sensing related variables in the topsoil organic carbon density of Liaohekou coastal wetlands, Northeast China ”
Dear authors,
The research paper has been revised properly. Just, add the units for BD and D on line 134.
Response: Thank you for your insightful comments. According to your comments, we have revised the manuscript. L138-139
We thank the reviewer 2 constructive comments, which significantly help improve our manuscript.

Reviewer 3 Report
Deara Editor and dear Authors,
Authors was makeing corrections for all suggestions.
The paper can be printeed.
Best regards,
Snežana Belanović Simić
Author Response
Response to reviewer 3 comments on on the manuscript remotesensing-1398759 “Prediction potential of remote sensing related variables in the topsoil organic carbon density of Liaohekou coastal wetlands, Northeast China ”
Dear Editor and dear Authors,
Authors was makeing corrections for all suggestions.
The paper can be printeed.
Response: Thank you for your careful and patient comments.
We thank the reviewer 3 constructive comments, which significantly help improve our manuscript.

This manuscript is a resubmission of an earlier submission. The following is a list of the peer review reports and author responses from that submission.